# Synthesizing realistic neural population activity patterns using Generative Adversarial Networks

**Manuel Molano-Mazon**[1,+]**, Arno Onken**[1,2]**, Eugenio Piasini**[1,3,*]**, Stefano Panzeri**[1,*]

[1]Laboratory of Neural Computation, Istituto Italiano di Tecnologia, 38068 Rovereto (TN), Italy
[2]University of Edinburgh, Edinburgh EH8 9AB, UK
[3]University of Pennsylvania, Philadelphia, PA 19104
[+]Corresponding author
[*]Equal contribution
`manuel.molano@iit.it, aonken@inf.ed.ac.uk, epiasini@sas.upenn.edu,`
`stefano.panzeri@iit.it`

## Abstract

The ability to synthesize realistic patterns of neural activity is crucial for studying neural information processing. Here we used the Generative Adversarial Networks (GANs) framework to simulate the concerted activity of a population of neurons. We adapted the Wasserstein-GAN variant to facilitate the generation of unconstrained neural population activity patterns while still benefiting from parameter sharing in the temporal domain. We demonstrate that our proposed GAN, which we termed Spike-GAN, generates spike trains that match accurately the first- and second-order statistics of datasets of tens of neurons and also approximates well their higher-order statistics. We applied Spike-GAN to a real dataset recorded from salamander retina and showed that it performs as well as state-of-the-art approaches based on the maximum entropy and the dichotomized Gaussian frameworks. Importantly, Spike-GAN does not require to specify *a priori* the statistics to be matched by the model, and so constitutes a more flexible method than these alternative approaches. Finally, we show how to exploit a trained Spike-GAN to construct 'importance maps' to detect the most relevant statistical structures present in a spike train. Spike-GAN provides a powerful, easy-to-use technique for generating realistic spiking neural activity and for describing the most relevant features of the large-scale neural population recordings studied in modern systems neuroscience.

## 1 Introduction

Understanding how to generate synthetic spike trains simulating the activity of a population of neurons is crucial for systems neuroscience. In computational neuroscience, important uses of faithfully generated spike trains include creating biologically consistent inputs needed for the simulation of realistic neural networks, generating large datasets to be used for the development and validation of new spike train analysis techniques, and estimating the probabilities of neural responses in order to extrapolate the information coding capacity of neurons beyond what can be computed from the neural data obtained experimentally (Ince et al., 2013; Moreno-Bote et al., 2014). In experimental systems neuroscience, the ability to develop models that produce realistic neural population patterns and that identify the key sets of features in these patterns is fundamental to disentangling the encoding strategies used by neurons for sensation or behavior (Panzeri et al., 2017) and to design closed-loop experiments (Kim et al., 2017) in which synthetic patterns, representing salient features of neural information, are fed to systems of electrical micro-stimulation (Tehovnik et al., 2006) or patterned light optogenetics (Panzeri et al., 2017; Bovetti & Fellin, 2015) for naturalistic intervention on neural circuits.

One successful way to generate realistic spike trains is that of using a bottom-up approach, focusing explicitly on replicating selected low-level aspects of spike trains statistics. Popular methods include

renewal processes (Stein (1965); Gerstner & Kistler (2002)), latent variable models (Macke et al., 2009; Lyamzin et al., 2010) and maximum entropy approaches (Tang et al., 2008; Schneidman et al., 2006; Savin & Tkačik, 2017), which typically model the spiking activity under the assumption that only first and second-order correlations play a relevant role in neural coding (but see Cayco-Gajic et al. (2015); Köster et al. (2014); Ohiorhenuan et al. (2010)). Other methods model spike train responses assuming linear stimulus selectivity and generating single trial spike trains using simple models of input-output neural nonlinearities and neural noise (Keat et al., 2001; Pillow et al., 2008; Lawhern et al., 2010). These methods have had a considerable success in modeling the activity of populations of neurons in response to sensory stimuli (Pillow et al., 2008). Nevertheless, these models are not completely general and may fail to faithfully represent spike trains in many situations. This is because neural variability changes wildly across different cortical areas (Maimon & Assad, 2006) due to the fact that responses, especially in higher-order areas and in behaving animals, have complex non-linear tuning to many parameters and are affected by many behavioral variables (e.g. the level of attention (Fries et al., 2001)).

An alternative approach is to apply deep-learning methods to model neural activity in response to a given set of stimuli using supervised learning techniques (McIntosh et al., 2016). The potential advantage of this type of approach is that it does not require to explicitly specify any aspect of the spike train statistics. However, applications of deep networks to generate faithful spike patterns have been rare. Here, we explore the applicability of the Generative Adversarial Networks (GANs) framework (Goodfellow et al., 2014) to this problem. Three aspects of GANs make this technique a good candidate to model neural activity. First, GANs are an unsupervised learning technique and therefore do not need labeled data (although they can make use of labels (Odena et al., 2016b; Chen et al., 2016)). This greatly increases the amount of neural data available to train them. Second, recently proposed modifications of the original GANs make them good at fitting distributions presenting multiple modes (Arjovsky et al., 2017; Gulrajani et al., 2017). This is an aspect that is crucial for neural data because the presentation of even a single stimulus can elicit very different spatio-temporal patterns of population activity (Churchland et al., 2007; Morcos & Harvey, 2016). We thus need a method that generates sharp realistic samples instead of producing samples that are a compromise between two modes (which is typical, for instance, of methods seeking to minimize the mean squared error between the desired output and the model's prediction (Goodfellow, 2016; Lotter et al., 2016)). Finally, using as their main building block deep neural networks, GANs inherit the capacity of scaling up to large amounts of data and therefore constitute a good candidate to model the ever growing datasets provided by experimental methods like chronic multi-electrode and optical recording techniques.

In the present work we extend the GAN framework to synthesize realistic neural activity. We adapt the recently proposed Wasserstein-GAN (WGAN) (Arjovsky et al., 2017) which has been proven to stabilize training, by modifying the network architecture to model invariance in the temporal dimension while keeping dense connectivity across the modeled neurons. We show that the proposed GAN, which we called Spike-GAN, is able to produce highly realistic spike trains matching the first and second-order statistics of a population of neurons. We further demonstrate the applicability of Spike-GAN by applying it to a real dataset recorded from the salamander retina (Marre et al., 2014) and comparing the activity patterns the model generates to those obtained with a maximum entropy model (Tkačik et al., 2014) and with a dichotomized Gaussian method (Lyamzin et al., 2010). Finally, we describe a new procedure to detect, in a given activity pattern, those spikes participating in a specific feature characteristic of the probability distribution underlying the training dataset.

## 2 METHODS

### 2.1 NETWORK ARCHITECTURE

We adapted the Generative Adversarial Networks described by Goodfellow et al. (2014) to produce samples that simulate the spiking activity of a population of $N$ neurons as binary vectors of length $T$ (spike trains; Fig. S3). In their original form, GANs proved to be difficult to train, prompting several subsequent studies that focused on making them more stable (Radford et al., 2015; Chintala et al., 2016). In the present work we used the Wasserstein-GAN variant described by Arjovsky et al. (2017). Wasserstein-GANs (WGAN) minimize the Earth-Mover (or Wasserstein-1) distance

(EM) between the original distribution $P_{\text{data}}$ and the distribution defined by the generator, $P_G$. Arjovsky et al. (2017) showed that the EM distance has desirable properties in terms of continuity and differentiability that ensure that the loss function provides a meaningful gradient at all stages of training, which boosts considerably its stability. A further improvement was later introduced by Gulrajani et al. (2017), who provided an alternative procedure to ensure that the critic is Lipschitz (via gradient penalization), which is required in the WGAN framework.

Here we adapted the WGAN-GP architecture (Gulrajani et al., 2017) to simulate realistic neural population activity patterns. Our samples are matrices of size $N \times T$, where $N$ is the number of neurons and $T$ the number of time bins, each bin usually corresponding to a few milliseconds (Fig. S3). Importantly, while samples present a high degree of invariance along the time dimension, they are usually not spatially structured (i.e. across neurons) and thus we cannot expect any invariance along the dimension spanning the different neurons. For this reason, in order to take advantage of the temporal invariance while being maximally agnostic about the neural correlation structure underlying the population activity, we modified a standard 1D-DCGAN (1 dimensional deep convolutional GAN) architecture (Radford et al., 2015) by transposing the samples so as to make the spatial dimension correspond to the channel dimension (Fig. 1). Therefore our proposed GAN can be seen as performing a *semi*-convolution, where the spatial dimension is densely connected while weights are shared across the temporal dimension thus improving training, efficiency and the interpretability of the trained networks.

The main modifications we have introduced to the WGAN-GP are:

1. The responses of different neurons are fed into different channels.

2. Following Chintala et al. (2016) we made all units LeakyReLU (the slope of the leak was set to 0.2) except for the last layer of the generator where we used sigmoid units.

3. The critic consists of two 1D convolutional layers with 256 and 512 features, respectively, followed by a linear layer (Fig. 1). The generator samples from a 128-dimension uniform distribution and its architecture is the mirror image of that of the critic.

4. To avoid the checkerboard issue described by Odena et al. (2016a), we divided all generator's fractional-strided convolutions (i.e. deconvolutions) into two separate steps: upsampling and convolving. The upsampling step is done using a nearest neighbor procedure, as suggested by Odena et al. (2016a).

We called the network described above Spike-GAN. As in Arjovsky et al. (2017), Spike-GAN was trained with mini-batch stochastic gradient descent (we used a mini-batch size of 64). All weights were initialized from a zero-centered normal distribution with standard deviation 0.02. We used the Adam optimizer (Kingma & Ba, 2014) with learning rate = 0.0001 and hyperparameters $\beta_1 = 0$ and $\beta_2 = 0.9$. The parameter $\lambda$, used for gradient penalization, was set to 10. The critic was updated 5 times for each generator update. All code and hyperparameters may be found at https://github.com/manuelmolano/Spike-GAN.

## 2.2 SPIKE TRAIN ANALYSIS

To compare the statistics of the generated samples to the ones contained in the ground truth dataset, we first discretized the continuously-valued samples produced by the generator and then, for each bin with activation $h$, we drew the final value from a Bernoulli distribution with probability $h$. Note that the last layer of the generator contains a sigmoid function and thus the $h$ values can be interpreted as probabilities.

We assessed the performance of the model by measuring several spike train statistics commonly used in neuroscience: 1) Average number of spikes (spike-count) per neuron. 2) Average time course, which corresponds to the probability of firing in each bin, divided by the bin duration (measured in seconds). 3) Covariance between pairs of neurons. 4) Lag-covariance between pairs of neurons: for each pair of neurons, we shift the activity of one of the neurons by one bin and compute the covariance between the resulting activities. This quantity thus indicates how strongly the activity of one of the neurons is related to the future activity of the other neuron. 5) Distribution of synchrony (or k-statistic), which corresponds to the probability $P_N(k)$ that $k$ out of the $N$ neurons spike at the same time. 6) Spike autocorrelogram, computed by counting, for each spike, the number of

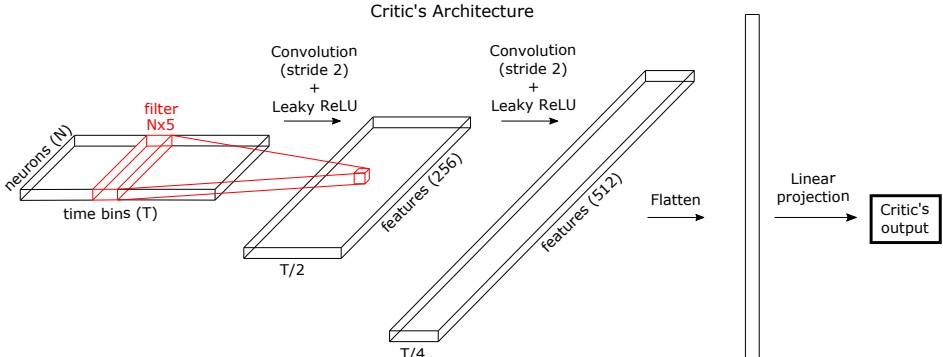

Figure 1: Critic's architecture. Samples are transposed so as to input the neurons' activities into different channels. The convolutional filters (red box) span all neurons but share weights across the time dimension. The critic consists of two 1D convolutional layers with 256 and 512 features. Stride=2; all units are LeakyReLU (slope=0.2). The architecture of the generator is the same as that of the critic, used in the opposite direction.

spikes preceding and following the given spike in a predefined time window. The obtained trace is normalized to the peak (which is by construction at 0 ms) and the peak is then zeroed in order to help comparisons.

## 3  RESULTS

### 3.1  FITTING THE STATISTICS OF SIMULATED SPIKE TRAINS

We first tested Spike-GAN with samples coming from the simulated activity of a population of 16 neurons whose firing probability followed a uniform distribution across the whole duration (T=128 ms) of the samples (bin size=1 ms, average firing rate around 100 Hz, Fig. 2D). In order to test whether Spike-GAN can approximate second-order statistics, the neurons' activities present two extra features that are commonly found in neural recordings. First, using the method described in Mikula & Niebur (2003), we introduced correlations between randomly selected pairs of neurons (8 pairs of correlated neurons; correlation coefficient values around 0.3). Second, we imposed a common form of temporal correlations arising from neuronal biophysics (*refractory period*): following an action potential, a neuron typically remains silent for a few milliseconds before it is able to spike again. This phenomenon has a clear effect on the spike autocorrelogram that shows a pronounced drop in the number of spikes present at less than 2 ms (see Fig. 2E). We trained Spike-GAN on 8192 samples for 500000 iterations (Fig. S4 shows the critic's loss function across training).

A representative sample produced by a trained Spike-GAN together with the resulting patterns (after binarizing the samples, see Section 2.2) is shown in Fig. 2A. Note that the sample (black traces) is mostly binary, with only a small fraction of bins having intermediate values between 0 and 1. We evaluated the performance of Spike-GAN by measuring several spike train statistics commonly used in neuroscience (see Section 2.2). For comparison, we also trained a generative adversarial network in which both the generator and the critic are a 4-layer multi-layer perceptron (MLP) and the number of units per layer is adjusted so both models present comparable numbers of trainable variables (490 units per layer which results in $\approx$ 3.5M trainable variables). As Fig. 2 shows, while both models fit fairly well the first three statistics (mean spike-count, covariances and k-statistics), the Spike-GAN's approximation of the features involving time (average time course, autocorrelogram and lag-covariance) is considerably better than that of the MLP GAN. This is most likely due to the weight sharing performed by Spike-GAN along the temporal dimension, that allows it to easily learn temporally invariant features.

In Supp. Section A.1 we further show that Spike-GAN is not only memorizing the samples present in the training dataset but it is able to effectively mimic their underlying distribution.

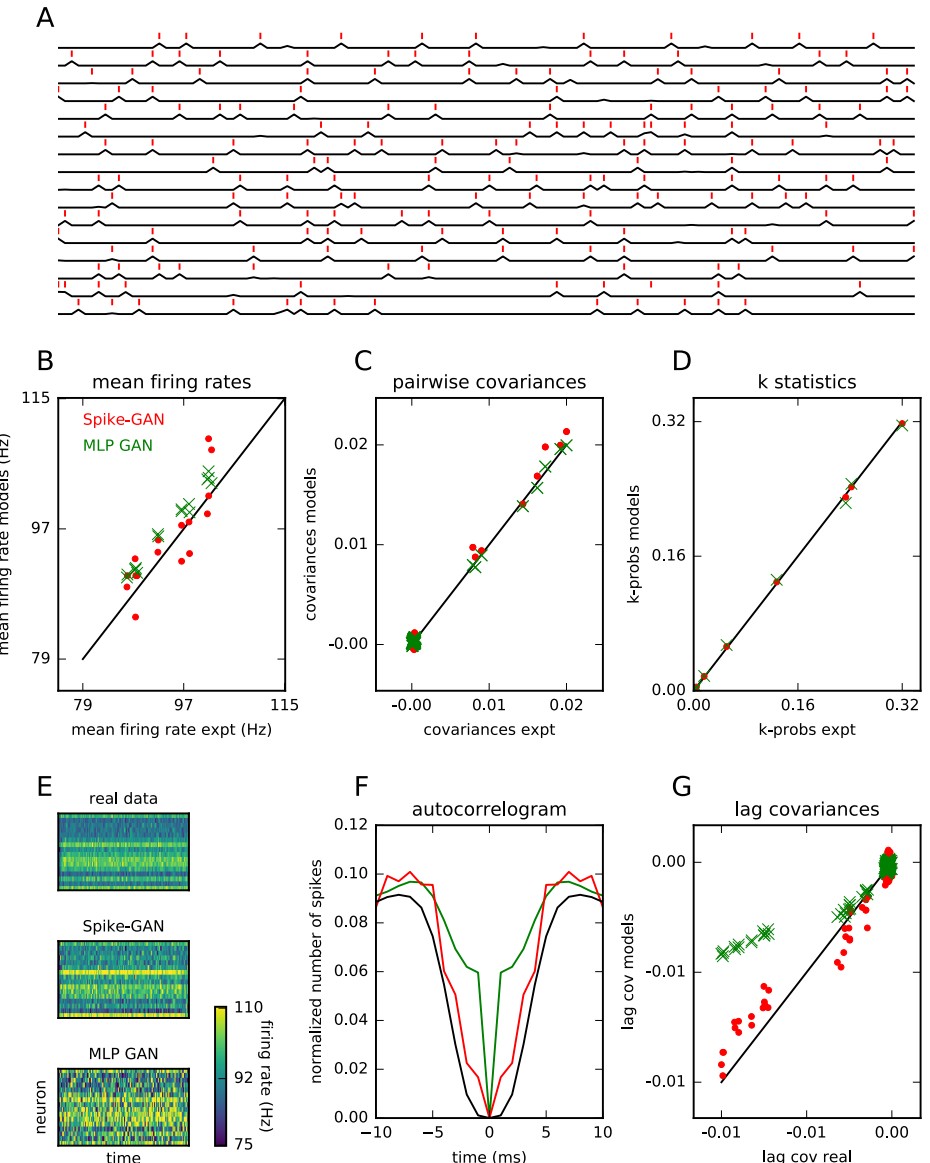

Figure 2: Fitting the statistics of simulated population activity patterns. A) Representative sample generated by Spike-GAN (black lines) and the resulting spike trains after binarizing (red lines). B-D) Fitting of the average spike-count, pairwise covariances and k-statistics done by Spike-GAN (red dots) and by a MLP GAN (green dots). Line indicates identity. E) Average time courses corresponding to the ground truth dataset and to the data obtained with Spike-GAN and the MLP GAN. F-G) Fitting of the autocorrelogram and the lag-covariances done by Spike-GAN (red line/dots) and a MLP GAN (green line/dots). Black line corresponds to the autocorrelogram resulting from the ground truth distribution.

## 3.2 Comparing to state-of-the-art methods

We next tested the Spike-GAN model on real recordings coming from the retinal ganglion cells (RGCs) of the salamander retina (Marre et al., 2014; Tkačik et al., 2014). The dataset contains the response of 160 RGCs to natural stimuli (297 repetitions of a 19-second movie clip of swimming fish and water plants in a fish tank) discretized into bins of 20 ms. We randomly selected 50 neurons out of the total 160 and partitioned their activity into non-overlapping samples of 640 ms (32 time bins) which yielded a total of 8817 training samples (using overlapping samples and thus increasing their number did not improve the results shown below). We obtained almost identical results for a different set of 50 randomly selected neurons (data not shown).

In order to provide a comparison between Spike-GAN and existing state-of-the-art methods, we fit the same dataset with a maximum entropy approach developed by Tkačik et al. (2014), the so-called k-pairwise model, and a dichotomized Gaussian method proposed by Lyamzin et al. (2010). Briefly, maximum entropy (MaxEnt) models provide a way of fitting a predefined set of statistics characterizing a probability distribution while being maximally agnostic about any other aspect of such distribution, i.e. maximizing the entropy of the probability distribution given the constraints in the statistics (Pressé et al., 2013). In neuroscience applications, the most common approach has been to design MaxEnt models fitting the first and second-order statistics, i.e. the average firing rates and pairwise correlations between neurons (Tang et al., 2008; Schneidman et al., 2006; Shlens et al., 2006). The k-pairwise model extends this approach to further constrain the activity of the neural population by fitting the k-statistics of the dataset of interest, which provides a measure of the neural population synchrony (see Section 2.2). Dichotomized Gaussian (DG) methods, on the other hand, model the neural activity by thresholding a correlated multivariate normal distribution with mean and covariance chosen such that the generated samples have the desired first- and second-order statistics. The method developed by Lyamzin et al. (2010) is an extension of previous approaches (see e.g. Macke et al. (2009)) in which signal and noise correlations are modeled separately. Importantly, unlike the k-pairwise model (and most MaxEnt models (Savin & Tkačik, 2017)), the DG model can fit temporal correlations.

We first checked for signs of overfitting by plotting, for each method, randomly selected generated samples together with their closest sample (in terms of L1 distance) in the training dataset. Although the generator in a GAN never 'sees' the training dataset directly but instead obtains information about the dataset only through the critic, it is still possible that the generator obtains enough information about the real samples to memorize them. Fig. S5 shows that this is not the case, with the closest samples in the training dataset being very different from the generated ones.

As shown in Fig. 3, all methods provide a good approximation of the average firing rate, the covariance and the k-statistics, but the fit performed by the MaxEnt (green dots) and the DG (blue pluses) models is somewhat tighter than that produced by Spike-GAN (red dots). This is not surprising, as these are the aspects of the population activity distribution that these models are specifically designed to fit. By contrast, Spike-GAN does remarkably well without any need for these statistical structures to be manually specified as features of the model.

As mentioned above, the k-pairwise model does not take into account the temporal dynamics of the population and therefore ignores well-known neural features that are very likely to play a relevant role in the processing of incoming information (e.g. refractory period, burst or lagged cross-correlation between pairs of neurons). Fig. 3 shows that both Spike-GAN and the DG model approximate well the ground truth autocorrelogram and lag-covariances while the k-pairwise model, as expected, entirely fails to do so.

Importantly, while its performance in terms of reproducing positive correlations is remarkable, the DG method struggles to approximate the statistics of neural activity associated with negative correlations (Lyamzin et al., 2010). Fig. S6 shows how the k-pairwise and the DG methods fit the dataset described in Fig. 2. As can be seen, the DG model, while matching perfectly the (positive) correlations between neurons, fails to approximate the negative correlations present in the autocorrelogram that are caused by the refractory period (Fig. S6E).

The above results demonstrate that Spike-GAN generates samples comparable to those produced by state-of-the-art methods without the need of defining *a priori* which statistical structures constitute important features of the probability distribution underlying the modeled dataset.

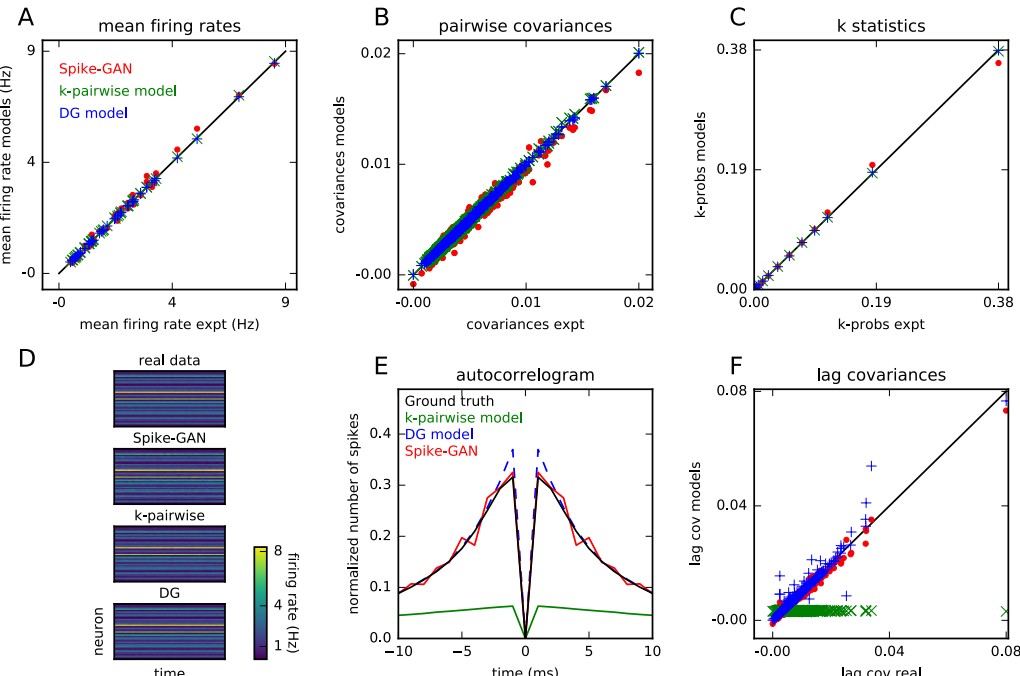

Figure 3: Fitting the statistics of real population activity patterns obtained in the retinal salamander. A-C) Fitting of the average spike-count, pairwise covariances and k-statistics done by Spike-GAN (red dots), the k-pairwise model (green crosses) and the DG model (blue pluses). Line indicates identity. D) Average time courses corresponding to the ground truth data and to the data obtained with Spike-GAN, the k-pairwise model and the DG model. E-F) Fitting of the autocorrelogram and the lag-covariances done by Spike-GAN (red line/dots), the k-pairwise model (green line/crosses) and the DG model (blue dashed line/pluses). Black line corresponds to the autocorrelogram resulting from the ground truth distribution.

### 3.3 USING THE TRAINED CRITIC TO INFER RELEVANT NEURAL FEATURES

We then investigated what a trained critic can tell us about the population activity patterns that compose the original dataset. In order to do so, we designed an alternative dataset in which neural samples contain stereotyped activation patterns each involving a small set of neurons (Fig. 4A). This type of activation patterns, also called packets, have been found in different brain areas and have been suggested to be fundamental for cortical coding, forming the basic symbols used by populations of neurons to process and communicate information about incoming stimuli (Luczak et al., 2015). Thus, besides being a good test for the capability of Spike-GAN to approximate more intricate statistical structures, analyzing simulated samples presenting packets constitutes an excellent way of demonstrating the applicability of the model to a highly relevant topic in neuroscience. We trained Spike-GAN on a dataset composed of neural patterns of 32 neurons by 64 ms that present four different packets involving non-overlapping sets of 8 neurons each (Fig. 4A). Importantly, only few neurons out of all the recorded ones typically participate in a given packet and, moreover, neurons are usually not sorted by the packet to which they belong. Therefore, real neural population activity is extremely difficult to interpret and packets are cluttered by many other 'noisy' spikes (Fig. 4B). In order to assess the applicability of Spike-GAN to real neuroscience experiments, we trained it on these type of realistic patterns of activity (see caption of Fig. 4 for more details on the simulated dataset and the training).

Visual inspection of the filters learned by the first layer of the critic suggests that Spike-GAN is able to learn the particular structure of the packets described above: many of the filters display spatial distributions that are ideally suited for packet detection (Fig. S7; note that filters have been sorted in the neurons' dimension to help visualization).

Recently, Zeiler & Fergus (2014) developed a procedure to investigate which aspects of a given sample are most relevant for a neural network. They proposed to systematically alter different parts of the input and evaluate the change each alteration produces in the output of different layers of the network. Here we have adapted this idea to investigate which are the most relevant features of a given neural activity pattern. We first compute the output produced by the critic for a real sample. Then, for a given neuron and a given temporal window of several milliseconds, we shuffle across time the spikes emitted by the neuron during that period of time and compute the output of the critic when using as input the altered sample. The absolute difference between the two outputs gives us an idea of how important is the structure of the spike train we have disrupted. We can then proceed in the same fashion for all neurons and for several time windows and obtain a map of the importance of each particular spike train emitted by each neuron (importance maps, see Fig. 4C, heatmaps).

To highlight the usefulness of the procedure explained above, we produced a separate dataset in which the same population of neurons encodes the information about a particular stimulus by emitting one of the packet types shown in Fig. 4A around 16 ms after the stimulus presentation [1]. Fig. 4C (gray scale panels) shows 5 representative example patterns (see also Fig. S8). The packets are highlighted for visualization, but it is clear that patterns containing packets are almost indistinguishable from those without them. Noticeably, the importance maps (heatmaps) are able to pinpoint the spikes belonging to a packet (note that this does not require re-training of Spike-GAN). Further, by averaging the importance maps across time and space, we can obtain unambiguous results regarding the relevance of each neuron and time period (Fig. 4D-E; in Fig. 4E the neurons presenting higher importance values are those participating in the packet).

The importance-map analysis thus constitutes a very useful procedure to detect the most relevant aspects of a given neural population activity pattern. In Fig. S2 we describe a potential application of the importance maps to the study of how a population of neurons encode the information about a given set of stimuli.

## 4 DISCUSSION

We explored the application of the Generative Adversarial Networks framework (Goodfellow et al., 2014) to synthesize neural responses that approximate the statistics of the activity patterns of a

---

[1]It has been shown that, in the sensory cortex, activity packets in response to external stimuli are very similar to those recorded when no stimulation is applied (Luczak et al., 2015).

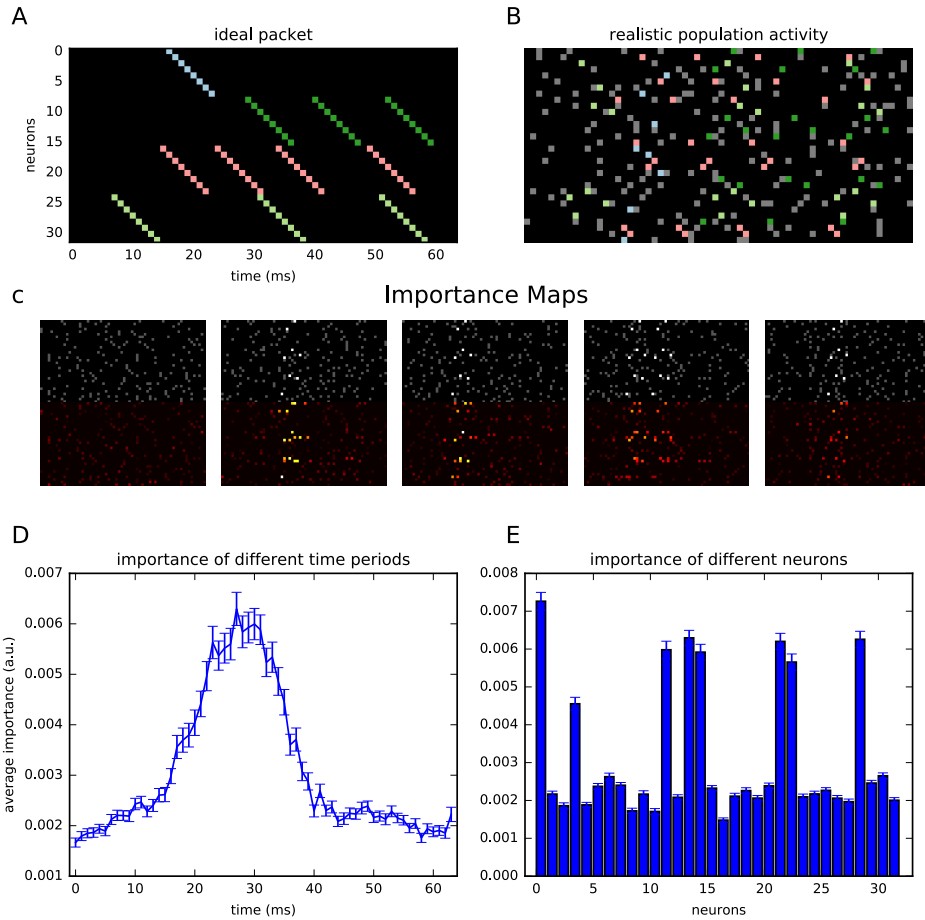

Figure 4: A) An example pattern showing the different packets highlighted with different colors and sorted to help visualization. The probability of each type of packet to occur was set to 0.1. Packets of the same type do not overlap in time. B) Realistic neural population pattern (gray spikes do not participate in any packet). C) Examples of activity patterns (grayscale panels) in which only one type of packet is usually present (one or two times) during a period of time from 16 to 32 ms. Packets are highlighted as white spikes. Heatmaps: importance maps showing the change that disrupting specific spikes has on the critic's output. Note that packet spikes normally show higher values. We used a sliding window of 8 ms (with a step size of 2 ms) to selectively shuffle the activity of each neuron at different time periods. The Spike-GAN used to obtain these importance maps was trained for 50000 iterations on 8192 samples. D) Average of 200 randomly selected importance maps across the neurons dimension, yielding importance as a function of time. E) Average of the same 200 randomly selected importance maps across the time dimension, yielding importance as a function of neurons. Errorbars correspond to standard error.

population of neurons. For this purpose, we put forward Spike-GAN, by adapting the WGAN variant proposed by Arjovsky et al. (2017) to allow sharing weights across time while maintaining a densely connected structure across neurons. We found that our method reproduced to an excellent approximation the spatio-temporal statistics of neural activity on which it was trained. Importantly, it does so without the need for these statistics to be handcrafted in advance, which avoids making *a priori* assumptions about which features of the external world make neurons fire.

Recently, Pandarinath et al. (2017) have proposed a deep learning method, LFADS (Latent Factor Analysis via Dynamical Systems), to model the activity of a population of neurons using a variational autoencoder (in which the encoder and decoder are recurrent neural networks). LFADS allows inferring the trial-by-trial population dynamics underlying the modeled spike train patterns and thus

can be seen as a complementary method to Spike-GAN, which does not explicitly provide the latent factors governing the response of the neurons. Regarding the application of the GANs framework to the field of neuroscience, Arakaki et al. (2017) proposed a GAN-based approach for fitting network models to experimental data consisting of a set of tuning curves extracted from a population of neurons. However, to the best of our knowledge our work is the first to use GANs to *directly* produce realistic neural patterns simulating the activity of populations of tenths of neurons.

Building on the work by Zeiler & Fergus (2014), we showed how to use Spike-GAN to visualize the particular features that characterize the training dataset. Specifically, Spike-GAN can be used to obtain importance maps that highlight the spikes that participate in generating activity motifs that are most salient in the spike trains. This can be useful for unsupervised identification of highly salient low-dimensional representations of neural activity, which can then be used to describe and interpret experimental results and discover the key units of neural information used for functions such as sensation and behavior.

A further and promising application of importance maps is that of designing realistic patterns of stimulation that can be used to perturb populations of neurons using electrical or optical neural stimulation techniques (Panzeri et al., 2017; Tehovnik et al., 2006; Emiliani et al., 2015). The ability of Spike-GAN to generate realistic neural activity including its temporal dynamics and to identify its most salient features suggests that it may become a very relevant tool to design perturbations. In Fig. S2 we provide a more detailed description of a potential application of Spike-GAN, in which importance maps may allow inferring the set of neurons participating in the encoding of the information about a given set of stimuli (Fig. S2F) and the spatio-temporal structure of the packets elicited by each stimulus (Fig. S2E).

We have compared Spike-GAN with two alternative methods based on the maximum entropy and the dichotomized Gaussian frameworks. These methods offer the possibility of computing the sample probabilities (MaxEnt model) and separately specifying the signal and noise correlations present in the generated samples (DG model). Spike-GAN does not have these features; nevertheless, it does have important advantages over the mentioned methods. First, Spike-GAN is more flexible than the MaxEnt and DG models, being able to fit any type of spatio-temporal structure present in the data. Further, it does not require making *a priori* assumptions about which statistical properties of a dataset are relevant and thus need to be matched. Finally, Spike-GAN is based on the deep neural network framework, and is therefore able to directly benefit from the engineering advances emerging in this rapidly-growing field. Conceivably, this will enable Spike-GAN, or methods derived from it, to make in the future better and better use of the datasets of ever increasing size that are produced by the experimental neuroscience community.

### ACKNOWLEDGMENTS

This work has received funding from the European Union's Horizon 2020 research and innovation programme under the Marie Sklodowska-Curie grant agreement No 699829 (ETIC) and under the Marie Sklodowska-Curie grant agreement No 659227 (STOMMAC). EP thanks Vittal Premachandran for discussions at the Brains, Minds and Machines summer course.

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

# A APPENDIX

## A.1 GENERATING NEW, REALISTIC PATTERNS OF NEURAL ACTIVITY

In this section, we investigated how well Spike-GAN fits the whole probability density function from which the population activity patterns present in the training dataset are drawn, following an approach inspired by Macke et al. (2009).

We started by producing a *ground truth* dataset ($2 \cdot 10^6$ samples) for a small-size problem (2 neurons x 12 time bins) so as to reduce the dimensionality of the samples and obtain a good approximation of the underlying probability density function. We will call the probabilities computed from the ground truth dataset *numerical probabilities*. We then obtained a second dataset ($2 \cdot 10^6$ samples) drawn from the same probability distribution as the *ground truth* dataset, which we will call *surrogate* dataset. This dataset will provide us with a reference for the results we obtain when comparing the distribution generated by Spike-GAN with the original ground truth distribution.

A third small dataset (*training* dataset, 8192 samples) coming from the same probability distribution as the *ground truth* and *surrogate* datasets, was used to train Spike-GAN. Finally, a $2 \cdot 10^6$-sample dataset (*generated* dataset) was obtained from the trained Spike-GAN.

The distributions corresponding to the three different datasets presented very similar entropies: the *ground truth* and *surrogate* datasets had both 14.6 bits while the *generated* dataset had 14.3 bits. Thus, the dataset generated by Spike-GAN was not affected by any evident mode collapse. We then plotted the sample probabilities with respect to both the *generated* and the *surrogate* dataset against the *numerical probabilities*. By comparing the densities in Fig. S1, we deduce that the surrogate probability distribution deviates from the ground truth distribution (the identity line) in the same way as the generated distribution does. Hence, this deviation can be attributed to finite sampling effects rather than poor performance of Spike-GAN.

Importantly, the percentage of samples generated by Spike-GAN that were originally present in the *training* dataset was $44\%$ ($45\%$ for the *surrogate* dataset). This implies that $56\%$ percent of the samples produced by Spike-GAN were generated *de novo*. Finally, the percentage of generated samples that had *numerical probability* equal to 0 was $3.2\%$, which is comparable to the $3.8\%$ of samples in the surrogate dataset that also had *numerical probability* 0.

Taken together, the above results strongly suggest that Spike-GAN has learned the probability distribution underlying the training dataset.

## A.2 USING IMPORTANCE MAPS IN A REAL EXPERIMENT

When recording large-scale spike trains it is difficult to make hypotheses about what are the key features of population activity that allow the animal to discriminate between the stimuli (for example, the patterns of firing rates or the pattern of response latencies of specific subsets of neurons). One possible use of Spike-GAN is to interpret the features of neural activity that are prominent in the importance maps as possible candidates for being units of information relevant for behavioral discrimination.

To illustrate the approach, we simulated a hypothetical experiment (Fig. S2A) in which we consider $N$ repetitions of a behavioural task, where a mouse has to discriminate two different stimuli (vertical/horizontal stripes). For each repetition of the task, we assume to record several patterns of neural activity such as those in Fig. 4 (main paper). By means of two-photon calcium imaging the activity of a population of V1 neurons in the visual cortex of the mouse is recorded in response to the two stimuli, which are associated with two distinct behavioral outcomes (e.g. drink from a left/right reward port (Alemi-Neissi et al., 2013)). The mouse is assumed to have been previously trained on the task. [Note that one possible difficulty in applying Spike-GAN to calcium imaging is that, unlike spikes directly extracted from electrophysiology experiments, two-photon imaging signals are not binary-valued. Nevertheless, to a first approximation, calcium signals can be binarized, and this has been shown not to constitute an obstacle to the study of sensory encoding or decision-making in certain preparations (Runyan et al., 2017).]

Area V1 is known to encode information about the stimulus orientation and therefore the two stimuli will evoke two distinct activity patterns (Fig. S2B). However, these patterns are usually difficult to

identify due to background activity and the lack of information about the ground truth correlation structure of the recorded population of neurons. Fig. S2C shows the actual 'recorded' population responses.

As described in Section 3.3, Spike-GAN can then be trained on the samples shown in Fig. S2C in order to compute the importance map associated to each sample (Fig. S2D). We can then approximate the structure of the original packets by thresholding and aligning (to the first spike above threshold) each importance map and averaging all aligned maps for each stimulus. Importantly, the approximated packets (gray maps) and the original ones (green pluses) are highly correlated (Fig. S2E; $r \geq 0.8$, yellow values) and the distribution of importance values corresponding to bins participating in a packet (those marked with a green plus) is clearly different from that corresponding to the rest of the bins (Fig. S10A). Furthermore, we can easily identify those neurons participating in the encoding of each stimulus by averaging the importance maps across time and trials for each stimulus (see caption to Fig. S2 for details). As shown in Fig. S2F, the neurons that (by design) participate in the encoding of a particular stimulus (indicated by red pluses and blue crosses for stim1 and stim2, respectively) are those presenting the highest importance values. Given the clear bimodal distribution of the importance values shown in Fig. S2F, we can easily set a threshold to select those neurons encoding each stimulus. The recall and precision values (Ting, 2011) corresponding to this threshold is also shown in Fig. S2F.

Finally, we have evaluated the importance maps in noisier scenarios, in which the spikes forming the packets present noise in their timing (Fig. S9) or in which the number of samples available to train Spike-GAN is smaller (Fig. S10). In both cases we have found that the importance maps analysis is able to provide highly valuable information about the neurons participating in the encoding of the stimulus information and their relative timing.

The features of neural activity individuated as important by the spike GAN importance maps may or may not be used by the animal to make a decision. To test this causally, one can replace, during a perceptual discrimination task, the presentation of a sensory stimulus with the presentation of a 2P-optogenetic activation pattern. Importantly, the experimenter can, in each stimulation trial, either keep or shuffle the features previously highlighted by the importance maps before imposing the pattern. Comparing, between the altered and original patterns, the behavioral report of the animal about which stimulus was presented can then be used to find out whether the information carried by each specific feature is used to produce a consistent behavioral percept. For instance, a stimulation protocol in which only part of the neurons participating in a particular feature is stimulated would provide information about the capacity of the mouse brain to 'fill in the gaps'. Alternatively, the time at which each neuron is stimulated could be altered in order to study the role spike timing plays in the encoding of the stimulus.

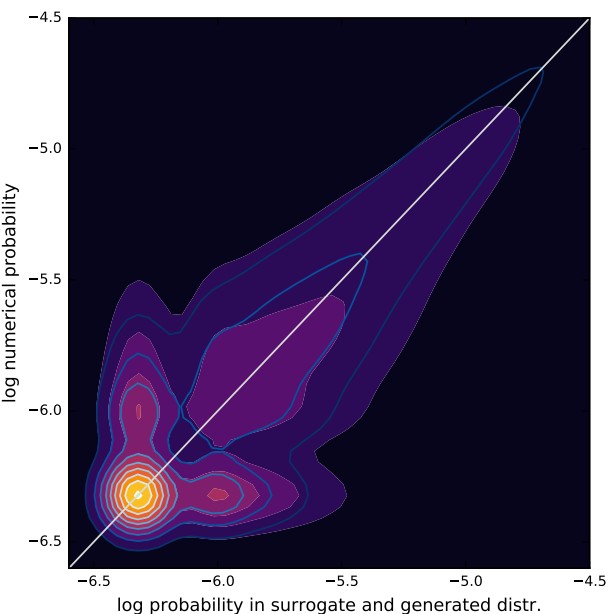

Figure S1: Fitting the whole probability distribution. *Numerical probabilities* obtained from the *ground truth* dataset (2 correlated neurons, samples duration=12 ms, firing rate≈160 Hz, correlation≈0.3, refractory period=2 ms) vs probabilities inferred from the *surrogate* and the *generated* datasets. Gray line is the identity. Blue tones: probabilities computed with respect to the *surrogate* dataset. Red tones: probabilities computed with respect to the *generated* dataset. Both distributions are obtained by kernel density estimation, with a 2D Gaussian kernel with bandwidth=0.1.

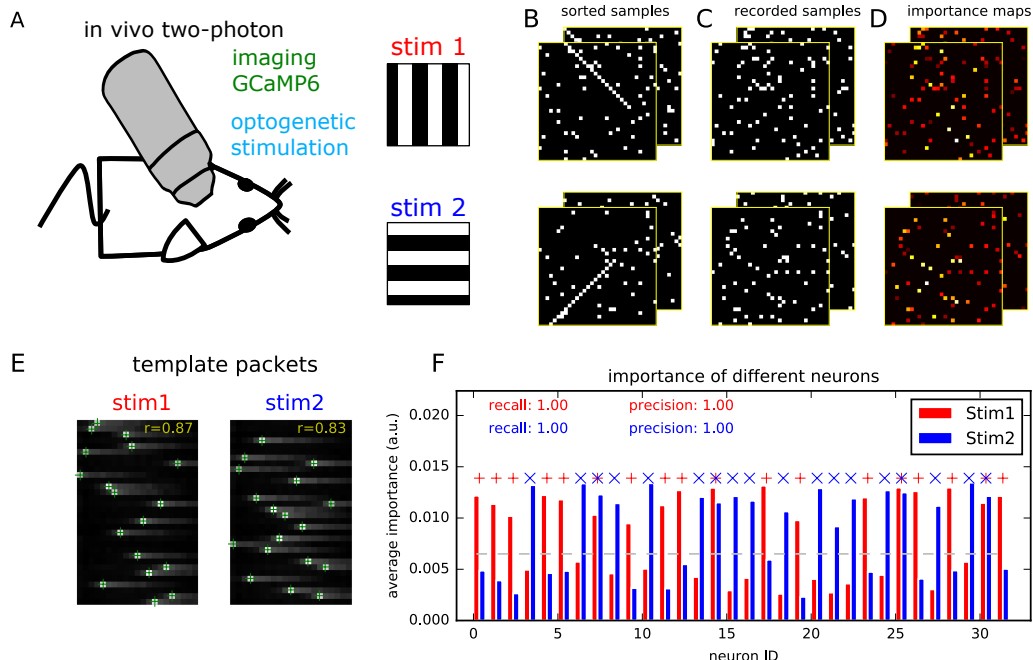

Figure S2: Using importance maps in a real experiment. A) Two-photon microscopy can be used for both imaging and manipulating the activity of a population of neurons. A trained mouse is presented with two different stimuli (stim1 and stim2) and makes a decision (e.g. drink from right/left reward port) accordingly. At the same time the response of a population of neurons in V1 is recorded. B) Simulated data presenting two different packets (18 neurons per packet, 4 of which participate in both packets; each stimulus presentation evokes a single packet) in response to the two different stimuli. The packets are only evident when the neurons are sorted. C) Actual recorded patterns of activity. D) Importance maps obtained after training Spike-GAN with 4096 samples like the one shown in panel C. E) Approximated packets are obtained by thresholding the importance maps (we used as threshold the median of the total distribution of importance values) and aligning each of them to the first spike above the threshold. The aligned maps are then averaged for each stimulus (gray maps). Green pluses show the structure of the ground truth packets (in yellow the correlation between the ground truth and the template packets). F) The importance of each neuron can be inferred by averaging importance maps across time and trials. Blue crosses and red pluses indicate the neurons that by design participate in the encoding of each stimulus, respectively. The computed importance for these neurons is more than three times that of the neurons that do not participate. Dashed gray line indicate the threshold used to select the participating neurons and compute the precision and recall values shown in red and blue for each stimulus, respectively.

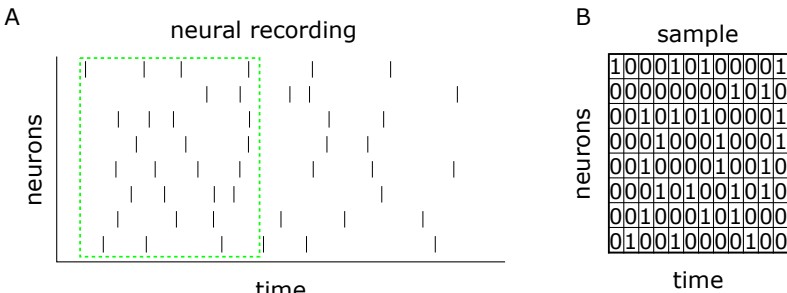

Figure S3: A) Example of neural recording in which the activity of 8 neurons is measured during an arbitrary time period. B) Example of a sample extracted from the activity shown in A (green box).

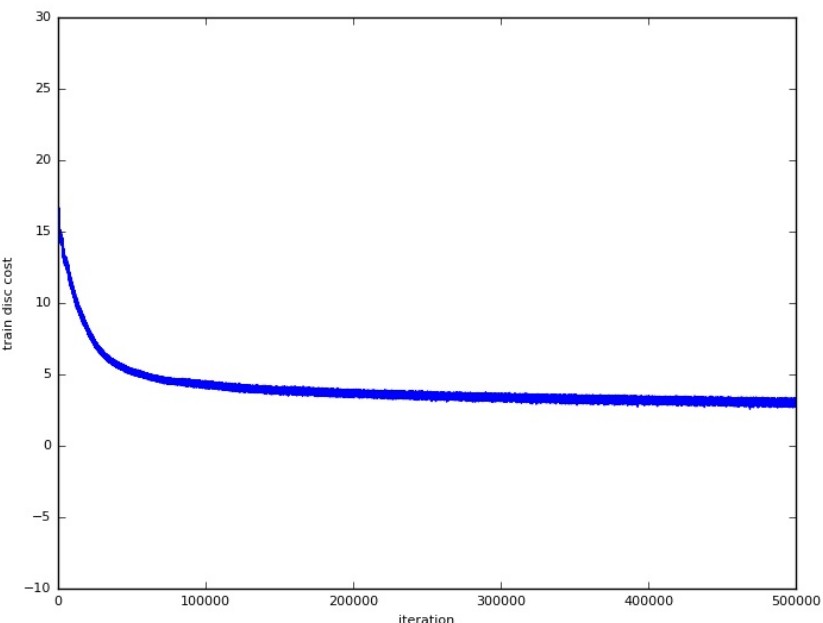

Figure S4: Negative critic loss corresponding to the training of Spike-GAN on samples coming from the simulated activity of a population of 16 neurons whose firing probability follows a uniform distribution across the whole duration (T=128 ms) of the samples (see Section 3.1).

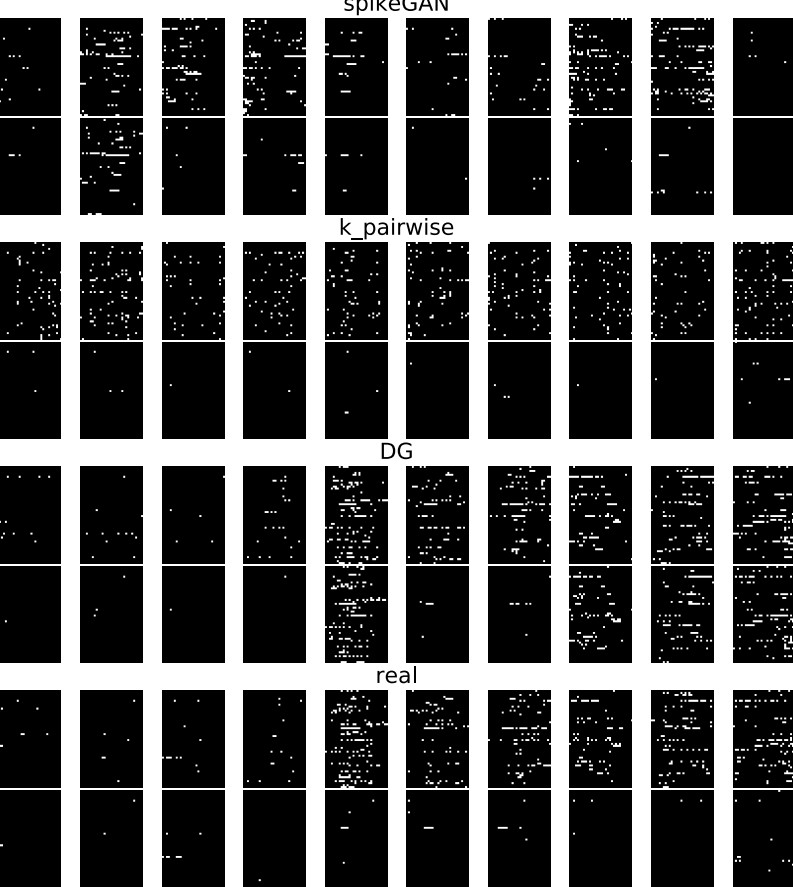

Figure S5: Ten generated samples are shown together with their closest sample in the training dataset for Spike-GAN, the k-pairwise and the DG method. Note that we are measuring sample similarity in terms of L1 distance. This implies that sometimes the closest sample is the one presenting no spikes since non matching spikes are penalized more. For comparison, 10 samples contained in the training dataset are shown together with their closest sample in the same training dataset (excluding the sample itself).

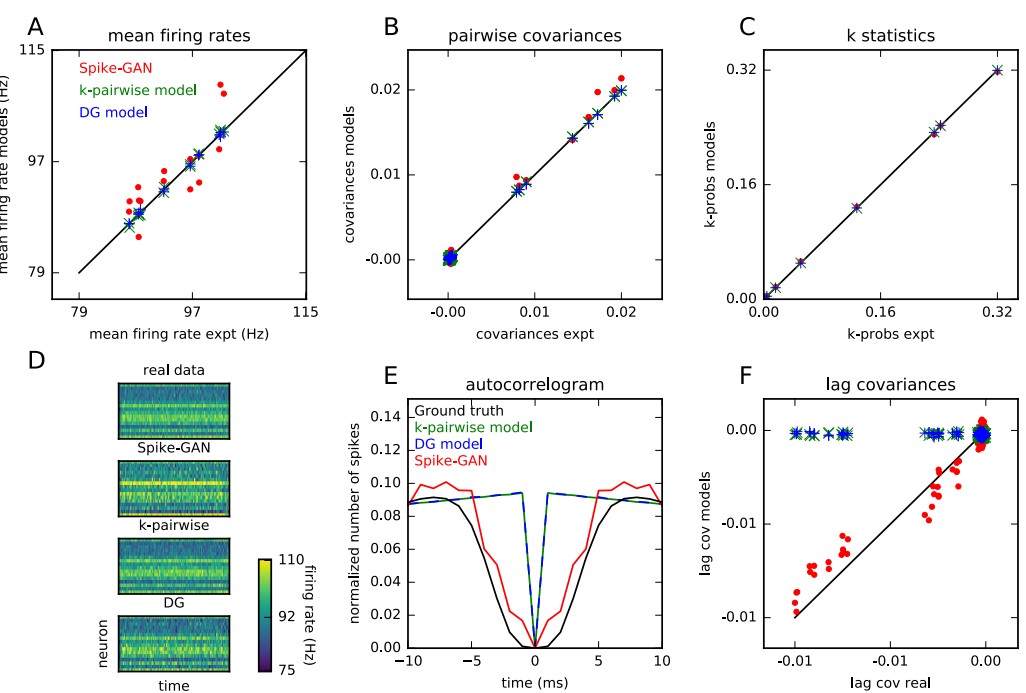

Figure S6: Comparison of Spike-GAN with the k-pairwise and the DG models in the presence of negative correlations (refractory period).

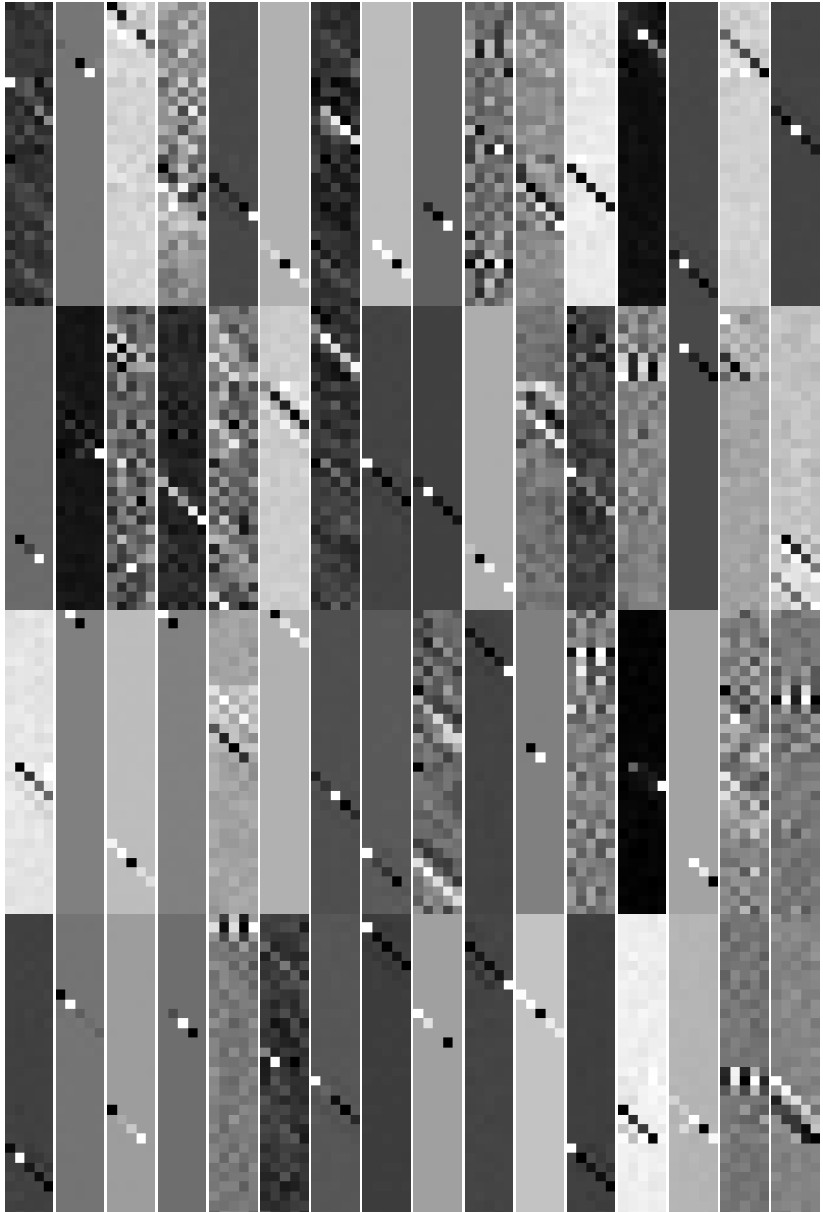

Figure S7: Filters learned by the first layer of Spike-GAN when trained on the dataset described in Section 3.3.

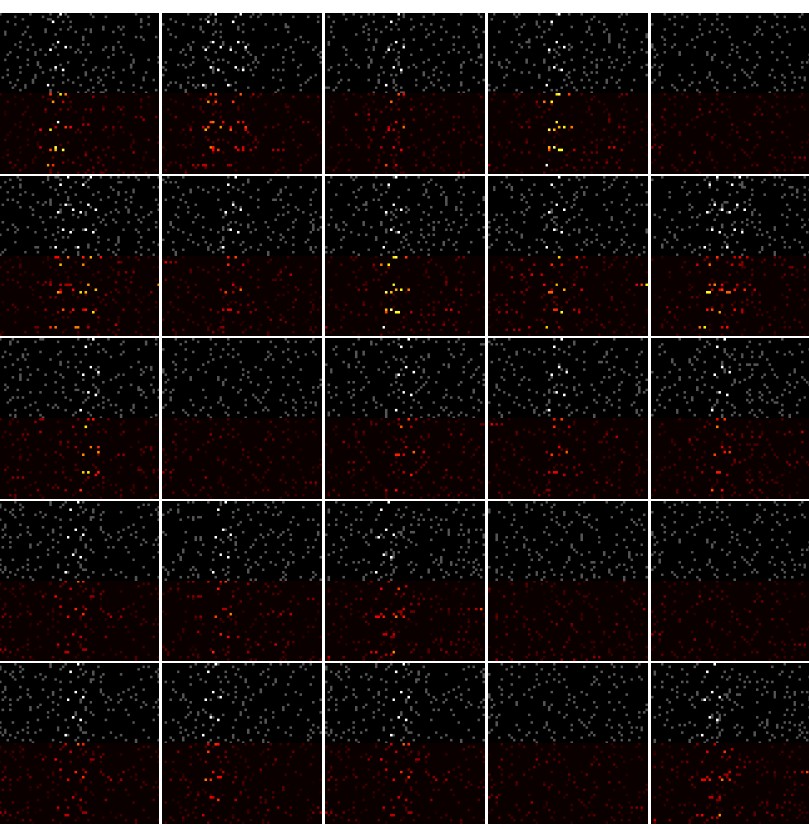

Figure S8: Randomly selected samples and their corresponding importance maps.

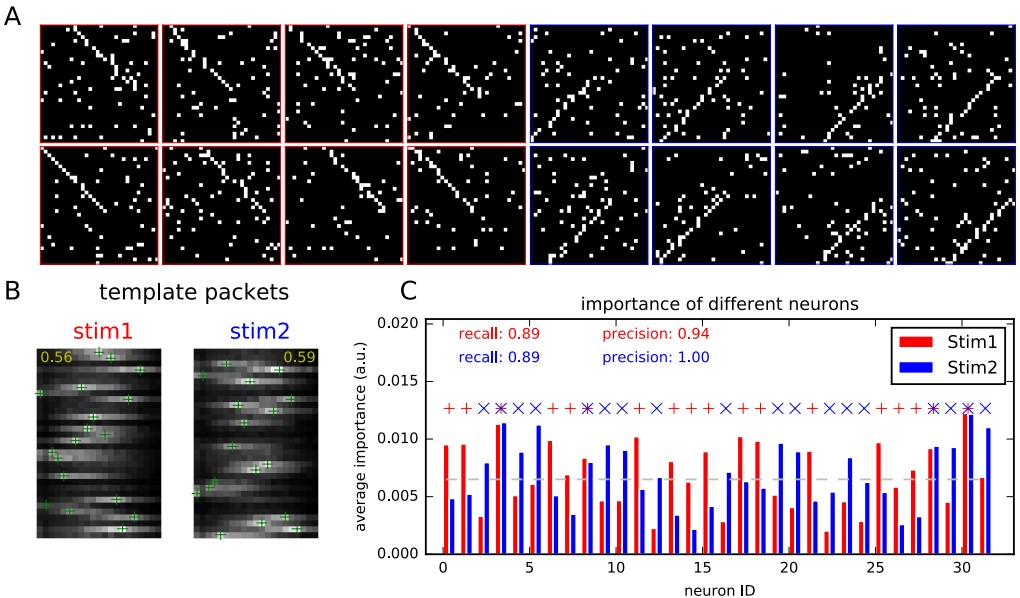

Figure S9: In order to test the approach discussed in Section A.2 with less reliable responses, we introduced noise in the packets in the form of discretized Gaussian noise (std=0.5) added to the time of each spike (panel A shows 8 randomly selected samples for each stimulus (red and blue boxes, respectively)). B) Inferred packets are, as expected, noisier but the correlation with the ground truth packets is still high (yellow values). Note that the correlation is computed between the inferred packet and the average of all *noisy* packets present in the training dataset, so as to take into account the inherent uncertainty of the responses. Green pluses show the structure of the ground truth, noise-free packets. F) Neurons importance value each stimulus (red and blue, respectively). Blue crosses and red pluses indicate the neurons that by design participate in the encoding of each stimulus, respectively. Dashed gray line indicate the threshold used to select the participating neurons and compute the precision and recall values shown in red and blue for each stimulus, respectively. Most of the relevant neurons are still being detected (recall=0.89) with just a slight decrease in the precision (precision $\geq 0.94$).

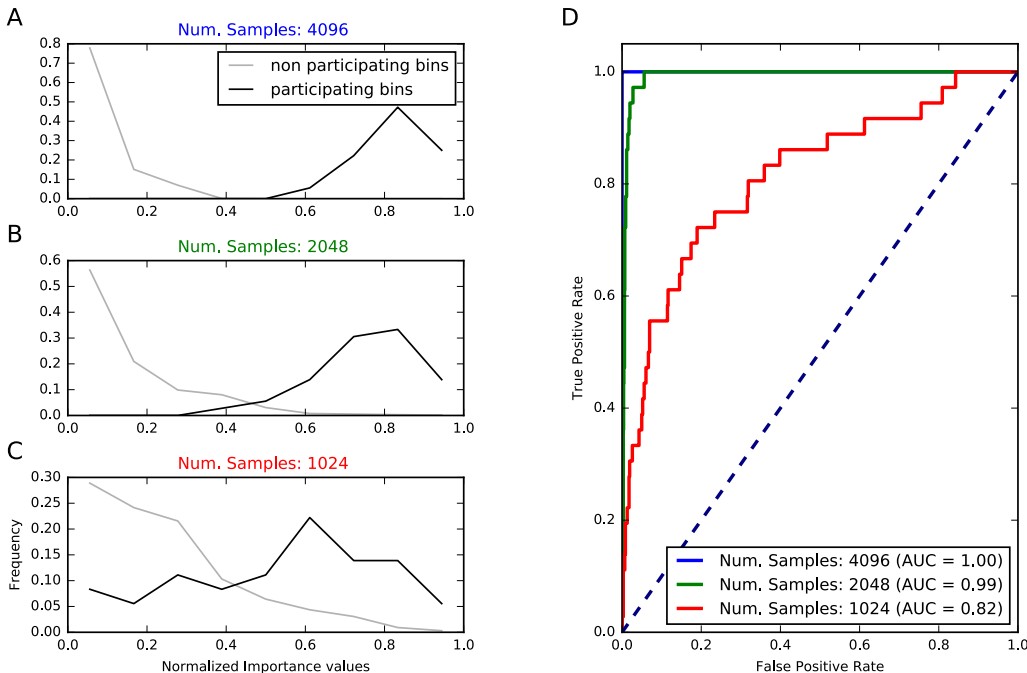

Figure S10: Approximated packets for different training dataset sizes. A) Distribution of the importance values (normalized to the maximum value) shown in Fig. S2E, for the bins forming the packets (i.e. bins marked with a green plus in Fig. S2E) (black line) and those not participating in the packets (gray line). The two distributions are clearly separated, as confirmed by the ROC curve in panel D (solid blue line). B-C) Same as in panel A in the case where 2048 and 1024 samples have been used to train Spike-GAN, respectively. The corresponding ROC curves are shown in green and red, respectively, in panel D.

