# OpenReview forum: "Synthesizing realistic neural population activity patterns using Generative Adversarial Networks"
_ICLR.cc/2018/Conference — Accept (Poster)_

### Official Review · AnonReviewer3 · 2017-11-27
**A new application of GANs: Simulating neural spike train recordings**

**Rating:** 8
**Confidence:** 5

**Review:**

[Summary of paper] The paper presents a method for simulating spike trains from populations of neurons which match empirically measured multi-neuron recordings. They set up a Wasserstein-GAN and train it on both synthetic and real multi-neuron recordings, using data from the salamander retina. They find that their method (Spike-GAN) can produce spike trains that visually look like the original data, and which have low-order statistics (firing rates, correlations, time-lagged-correlations, total sum of population activity) which matches those of the original data. They emphasize that their network architecture is 'semi-convolutional', i.e. convolutional in time but not across neurons. Finally, they suggest a way to analyse the fitted networks in order to gain insights into what the 'relevant' neural features are, and illustrate it on synthetic data into which they embedded these features.

[Originality] This paper falls into the category of papers that do a next obvious thing ("GANs have not been applied to population spike trains yet"), and which do it pretty well: If one wants to create simulated neural activity data which matches experimentally observed one, then this method indeed seems to do that. As far as I know, this would be the first peer-reviewed application of GANs to multi-neuron recordings of neural data (but see https://arxiv.org/abs/1707.04582 for an arxiv paper, not cited here-- should be discussed at least).  On a technical level, there is very little to no innovation here -- while the authors emphasise their 'semi-convolutional' network architecture, this is obviously the right architecture to use for multivariate time-series data, and in itself not a big technical novel. Therefore, the paper should really be evaluate as an `application' paper, and be assessed in terms of i) how important the application is, ii) how clearly it is presented, and iii) how convincing the results are relative to state of the art.

i) [Importance of problem, potential significance] Finding statistical models for modelling and simulating population spike trains is a topic which is extensively studied in computational neuroscience, predominantly using  model-based approaches using MaxEnt models, GLMs or latent variable models. These models are typically simple and restricted, and certainly fall short of capturing the full complexity of neural data. Thus, better, and more flexible solutions for this problem would certainly be very welcome, and have an immediate impact in this community.  However, I think that the approach based on GANs actually has two shortcomings which are not stated by the authors, and which possibly limit the impact of the method: First, statistical models of neural spike trains are often used to compute probabilities e.g. for decoding analyses— this is difficult or impossible for GANs. Second, one most often does not want to simulate data which match a specific recording, but rather which have specified statistics (e.g. firing rates and correlations)— the method here is based on fitting a particular data-set, and it is actually unclear to me when that will be useful.

ii) [Clarity] The methods are presented and explained clearly and cleanly. In my view, too much emphasis is given to highlighting the ‘semi-convolutional’ network, and, conversely, practical issues (exact architectures, cost of training) should be explained more clearly, possibly in an appendix. Similarly, the method would benefit from the authors releasing their code.

iii) [Quality, advance over previous methods] The authors discuss several methods for simulating spike trains in the introduction. In their empirical comparisons, however, they completely focus on a particular model-class (maximum entropy models, ME) which they label being the ‘state-of-the-art’. This label is misleading— ME models are but one of several approaches to modelling neural spike trains, with different models having different advantages and limitations (there is no benchmark which can be used to rank them...). In particular, the only ‘gain’ of the GAN over ME  models in the results comes from their ability of the GAN to match temporal statistics. Given that the ME models used by the authors are blind to temporal correlations, this is, of course (and as pointed out by the authors) hardly surprising. How does the GAN approach fair against alternative models which do take temporal statistics into account, e.g. GLMs, or simple moment-based method e.g. Krumin et al 2009, Lyamzin 2010, Gutnisky et al 2010— setting these up would be simple, and it would provide a non-trivial baseline for the ability of spike-GAN to outperform at least these models? While it true that GANs are much more expressive than the model-based approaches used in neuroscience, a clear demonstration would have been useful.

Minor comments:
  - p.3: The abbreviation “1D-DCGAN” is never spelled out.
  - p.3: The architecture of Spike-GAN is never explicitely given.
  - p.3: (Sec. 2.2) Statistic 2) “average time course across activity patterns” is unclear to me -- how does one select the activity patterns over which to average? Moreover, later figures do not seem to use this statistic.
  - p.4: “introduced correlations between randomly selected pairs” -- How many such pairs were formed?
  - p.7 (just above Discussion) At the beginning of this section, and for Figs. 4A,B, the texts suggests that packets fire spontaneously with a given probability. For Figs. 4C-E, a particular packet responds to a particular input. Is then the neuron population used in these figures different from the one in Figs. 4A,B? How did the authors ensure that a particular set of neurons respond to their stimulus as a packet? What stimulus did they use?
  - p.8 (Fig. 4E) Are the eight neurons with higher importance those corresponding to the packet? This is insinuated but not stated.
  - p.12 (Appendix A)
    + The authors do not mention how they produced their “ground truth” data. (What was its firing rate? Did it include correlations? A refractory period?)
    + Generating samples from the trained Spike-GAN is ostensibly cheap. Hence it is unclear why the authors did not  produce a large enough number of samples in order to obtain a 'numerical probability', just as they did for the ground truth data?
    + Fig. S1B: The figure shows that every sample has the same empirical frequency. This indicates more a lack of statistical power rather than any correspondence between the theoretical and empirical probabilities. This undermines the argument in the second paragraph of p.12. In the other hand, if the authors did approximate numerical probabilities for the Spike-GAN, this argument would no longer be required.
  - p.13 Fig. S1A,B: the abscissas mention “frequency”, while the ordinates mention “probability”
  - p.25 Fig. S4: This figure suggests that the first layer of the Spike-GAN critic sometimes recognizes the packet patterns in the data. However, to know whether this is true, we would need to compare this to a representation of the neurons reordered in the same way and identified by packet. I.e. one expects something something like figure like Fig. 4A, with the packets lining up with the recovered filters when neurons are ordered the same way.

---

> ### Author Response · Authors · 2017-12-21
> **Rebuttal**
>
> Thank you very much for your comments and suggestions. We have tried to address them in the revised manuscript (see also reply to reviewers 1 and 2). Briefly:
>
> We have removed the reference to the ‘semi-convolutional’ architecture from the tittle and abstract and only mention it in the methods (Section 2.1, second paragraph, last four lines; see also reply to reviewer 1).
>
> We thank you for the reference Arakaki et al. We now mention it in the Discussion section (second paragraph).
>
> We have modified Fig. 1 in order to provide all details about the architecture of Spike-GAN.
>
> We have now compared Spike-GAN with a method that does take into account temporal statistics (Lyamzin et al. 2010) (Section 3.2). We show in Fig. 3 that this method fits very well the statistics of the retinal data. However, as the authors mention in their paper, it struggles to fit negative correlations as the ones shown in Fig. 1 (panel F) that are due to the refractory period (Fig. S6). This would constitute an important shortcoming if e.g. one wants to fit the activity of a neural population including inhibitory neurons, which will be most likely anti-correlated with other neurons.
>
> We now comment in the Discussion section (last paragraph) the pros and cons of Spike-GAN in comparison to the MaxEnt and DG models.
>
> The abbreviation “1D-DCGAN” is now spelled out (Section 2.1, second paragraph).
>
> Average time courses are the probability of firing in each bin, divided by the bin duration (measured in seconds). We have tried to clarify the explanation in Section 2.2 (second paragraph). This statistic is shown in Figs. 2E and 3D.
>
> In Fig.2 there were 8 pairs of correlated neurons. We have now made this information explicit in the text (Section 3.1, first paragraph).
>
> The idea behind the data shown in 4C-E was to clearly show how Spike-GAN could be used to identify the packets. Initially Spike-GAN is trained with a dataset in which samples show all packets (Fig. 4A) but disordered and cluttered by background spikes (Fig. 4B). Then we investigate the capacity of Spike-GAN to detect those packets in a separate simulated dataset in which the neural population responds to a hypothetical stimulus with only one of the packets (Fig. 4C). We have now tried to clarify the text explaining Fig.4C-E (Section 3.3, third paragraph). We have also added a new supp. figure (Fig. S2) showing a potential application of the importance maps (see also the reply to the other two reviewers).
>
> We now explicitly state that the eight neurons with higher importance are those corresponding to the packet (Section 3.3, third paragraph).
>
> As you suggested we obtained a larger dataset from Spike-GAN for Fig. S1 and compare the probabilities inferred from it with the numerical probabilities. We have thus simplified Section A1 and Fig. S1, since we believe the new panel A is enough to show that Spike-GAN is learning the underlying probability distribution. We have further computed the entropies of the ground truth and generated distributions (as suggested by reviewer 1), to check for the possibility of Spike-GAN producing low entropy samples. Finally we now specify the parameters used for the data shown in Fig. S1 (see caption) and changed the label in the abscissas.
>
> Fig. S4 (now Fig. S7) was mainly shown to indicate that Spike-GAN had learned the packet structure. We realized this was not clear from the text so we have modified it (Section 3.3, first paragraph, last four lines).
>
> We will make the code available on GitHub upon acceptance of the paper.
>
> Finally, the main paper got somewhat longer than 8 pages (the recommended length for ICLR submissions), but if so advised by the reviewers we would shorten the paper by moving Fig. 1 to the supplementary material.
>
> Thank you again for these very useful comments. Please let us know if you have further comments/doubts.

---

### Official Review · AnonReviewer1 · 2017-11-27
**Some potentially interesting ideas, but many weaknesses**

**Rating:** 4
**Confidence:** 4

**Review:**

Summary:

The paper proposes to use GANs for synthesizing realistic neural activity patterns. Learning generative models of neural population activity is an important problem in computational neuroscience. The authors apply an established method (WGAN) to a new context (neuroscience). The work does not present a technical advance in itself, but it could be a valuable contribution if it created novel insights in neuroscience. Unfortunately, I do not see any such insights and am therefore not sure about the value of the paper.



Pros:

- Using GANs to synthesize neural activity patterns is novel (to my knowledge).

- Using GANs allows learning the crucial statistical patterns from data, which is more flexible than MaxEnt modeling, where one needs to define the statistics to be matched in advance.

- Using the critic to learn something about what are the crucial population activity patterns is an interesting idea and could be a valuable contribution.



Cons:

- Important architecture details missing: filter sizes, strides.

- The only demonstrated advantage of the proposed approach over MaxEnt models is that it models temporal correlations. However, this difference has nothing to do with MaxEnt vs. GAN. A 64d MaxEnt model does not care whether you’re modeling 64 neurons in a single time bin or 8 neurons in 8 subsequent time bins. Thus, the comparison in Fig. 3E,F is not apples to apples. An apples-to-apples comparison would be to use a MaxEnt model that includes multiple time lags (if that’s infeasible with 16 neurons x 32 bins, use a smaller model). Given that the MaxEnt model does a better job at modeling neuron-to-neuron correlations, I would expect it to also outperform the GAN at modeling temporal correlations. There may well be a scalability issue of the MaxEnt model to large populations and long time windows, but that doesn’t seem to be the current line of argument.

- GANs have well-known problems like mode collapse and low entropy of samples. Given the small amount of training examples (<10k) and large number of model parameters (3.5M), this issue is particularly worrisome. The authors do not address this issue, neither qualitatively nor quantitatively, although both would be possible:

  a) A quantitative approach would be to look at the entropy of the data, the MaxEnt model and the GAN samples. Given the binary and relatively low-dimensional nature of the observations, this may be feasible (in contrast to image data). One would potentially have to look at shorter segments and smaller subpopulations of neurons, where entropy estimation is feasible given the available amount of data, but it’s definitely doable

  b) Qualitative approaches include the typical one of showing the closest training example for each sample.

- The novel idea of using the critic to learn something about the crucial population activity patterns is not fleshed out at all. I think this aspect of the paper could be a valuable contribution if the authors focused on it, studied it in detail and provided convincing evidence that it can be useful in practice (or, even better, actually produced some novel insight).

  a) Visualizing the filters learned by the critic isn’t really useful in practice, since the authors used their ground truth knowledge to sort the neurons. In practice, the (unsorted) filters will look just as uninterpretable as the (unsorted) population activity they show.

  b) Detection of the ‘packets’ via importance maps is an interesting idea to find potential temporal codes without explicitly prescribing their hypothesized structure. Unfortunately, the idea is not really fleshed out or studied in any detail. In particular, it’s not clear whether it would still work in a less extreme scenario (all eight neurons fire in exact sequence).

- Poor comparison to state of the art. MaxEnt model is the only alternative approach tested. However, it is not clear that MaxEnt models are the state of the art. Latent variable models (e.g. Make et al. 2011) or more recent approaches based on autoencoders (Pandarinath et al. 2017; https://doi.org/10.1101/152884) are just among a few notable alternatives that the authors ignore.



Questions:

- Can sequences of arbitrary length be generated or is it fixed to 32 samples? If the latter, then how do envision the algorithm’s use in real-world scenarios that you propose such as microstimulation?



Minor:

- There is nothing “semicomvolutional” here. Just like images are multi-channel (color channels) 2D (x, y) observation, here the observations are multi-channel (neurons) 1D (time) observations.

- Fig. 3E is not time (ms), but time (x20 ms bins). Potentially the firing rates are also off by a factor of 20?

- Why partition the training data into non-overlapping segments? Seems like a waste of training data not to use all possible temporal crops.

---

> ### Author Response · Authors · 2017-12-21
> **Rebuttal**
>
> Thank you very much for this detailed review. Below, we reply to your comments and questions:
>
> We modified Fig.1 to show the whole architecture of the critic (the generator is just the mirror image). Filters sizes, strides, etc are provided in the figure as well. We hope this new figure provides all the information that was missing in the previous version of the paper.
>
> We now compare the method to an alternative approach based on the dichotomized Gaussian (DG) framework (a latent variable model) (Section 3.2). In particular, we have applied an extension of the method developed by Macke et al. 2009 which was suggested by reviewer 3 (Lyamzin et al. 2010). Importantly, this alternative method takes into account the temporal structure of the retinal dataset and fits to a great extent all considered statistics (new Fig. 3). However, as the authors of the paper mention, the method fails to fit negative correlations and produces a flat autocorrelogram when spike trains present a refractory period (Fig. S6). We thus believe that the flexibility of Spike-GAN in fitting all kinds of spatio-temporal structures commonly found in real data makes it an attractive alternative to more constrained methods like the MaxEnt and the DG methods.
>
> To check for signs of mode collapse or low entropy samples we did the following:
> 1)	In Section A1 and Fig. S1 we compare, for a relatively low dimensional dataset, the entropies of the ground truth distribution to that of the distribution generated by Spike-GAN.
> 2)	We have included a figure showing, for each of the considered methods (Spike-GAN, k-pairwise and DG) 10 generated samples with their closest ones in the retinal dataset (see Section 3.2, third paragraph on page 6 and Fig. S5).
>
> Please see the response to reviewer 2 regarding the application of the importance maps to investigate the strategies used by neural populations to encode and transmit the information about a set of stimuli. In the example we describe in Section A2 and Fig. S2, we provide a possible way in which importance maps could be used in combination with a microstimulation technique such as optogenetics. Furthermore, using the same example, we have tested the importance maps analysis in the presence of noise affecting the packets (Fig. S9) and found that the procedure can still provide useful information about the neurons encoding each presented stimulus and their relative timing.
>
> The length of the patterns generated by Spike-GAN is bounded from above by the length of the data samples that were used for training the network. Nevertheless, apart from considerations of computational cost and data availability, there are no a priori limits on the duration of the training samples. Hence, if sequences of a specific length are needed for an experimental protocol, they can be generated by training a suitably-sized Spike-GAN. Alternatively, multiple shorter samples could be generated with Spike-GAN and successively stitched together (even though this would, of course, fail to capture any long-range correlations spanning across multiple samples).
>
> Visualizing the filters learnt by the first layer of the critic is useful to evaluate to which extent the network has learnt about the packets and that is the main reason we added this figure (it is true that without knowing the ground truth structure, they are of less help). We have modified the text to make this clearer (Section 3.3, first paragraph, last four lines).
>
> Although we agree that there is nothing technically challenging in the “Semi-convolutional” Spike-GAN architecture, we deemed useful to give a name to this approach since we feel many people in neuroscience (outside the machine learning community) associate convNets with its application to images and do not consider applying it to neural recordings. Nevertheless, we have reduced the emphasis on the semi-convolutional operation (we now only mention it in the methods (Section 2.1, second paragraph, last four lines)).
>
>  We have fixed the typo of Fig. 3E.
>
> Partitions were made non-overlapping to avoid redundancy in the training dataset, but it is true that using all possible temporal windows would be a trivial way of augmenting the data. We have tested this alternative approach with the retinal data (Fig. 3) and did not see any improvement on the fitting of the data. However, we agree that this alternative way of building the training dataset could be advantageous in cases when the number of samples is a more critical factor. We mention this in the first paragraph of Section 3.2.
>
> Finally, the main paper got somewhat longer than 8 pages (the recommended length for ICLR submissions), but if so advised by the reviewers we would shorten the paper by moving Fig. 1 to the supplementary material.
>
> Thank you again for this very useful feedback which helped us to considerably improve the quality of our paper. Please let us know if you have any other comment or question.

---

### Official Review · AnonReviewer2 · 2017-11-30
**Interesting idea, but applicability unclear**

**Rating:** 6
**Confidence:** 3

**Review:**

The paper applies the GAN framework to learn a generative model of spike trains.  The generated spike trains are compared to traditional model fitting methods, showing comparable or superior ability to capture statistical properties of real population  activity.

This seems like an interesting exercise, but it’s unclear what it contributes to our understanding of neural circuits in the brain.  The advantage of structured models is that they potentially correspond to underlying mechanisms and can provide insight.  The authors point to the superior ability to capture temporal structure, but this does not seem like a fundamental limitation of traditional approaches.

The potential applicability of this approach is alluded to in this statement toward the end of the paper:

“...be used to describe and interpret experimental results and discover the key units of neural information used for functions such as sensation and behavior.”

It is left for the reader to connect the dots here and figure out how this might be done.  It would be helpful if the authors could at least sketch out a path by which this could be done with this approach.

Perhaps the most compelling application is to perturbing neural activity, or intervening to inject specific activity patterns into the brain.

---

> ### Author Response · Authors · 2017-12-21
> **Rebuttal**
>
> Thank you for your helpful comments. We addressed your request to sketch out a path by which Spike-GAN could be used to discover the key features of population activity that mediate behavior. We comment on this in the Discussion section (third and fourth paragraph). We also elaborate in more detail in a new supplementary section (Section A2, with Fig. S2), where we describe an experimental paradigm (involving techniques nowadays used by many laboratories), which could greatly benefit from the importance maps produced by Spike-GAN.
>
> Briefly, the importance maps obtained from Spike-GAN allow inferring the set of neurons participating in the encoding of the stimulus information (Fig. S2F) and the spatio-temporal structure of the packets elicited by each stimulus (Fig. S2E). Inferred packets could then be altered in a meaningful and precise way and then applied to the population of neurons using interventional techniques such as 2P optogenetics. This would allow to causally test still unanswered questions about the way populations of neurons encode and transmit information – for instance, the role that spike timing plays in the encoding of stimulus information.
>
> Finally, the main paper got somewhat longer than 8 pages (the recommended length for ICLR submissions), but if so advised by the reviewers we would shorten the paper by moving Fig. 1 to the supplementary material.
>
> We hope that Section A2 and Fig. S2 are clear and address your concern about the contribution of our work to the investigation of how neural populations in the brain process and communicate the information they receive. Please let us know if you have further comments or questions.

---

### Author Response · Authors · 2017-11-01
**Corrections**

We provide below a few comments aiming at correcting/clarifying some sentences:

* First sentence in Section 2.1. should read: We adapted the Generative Adversarial Networks described by Goodfellow et al. (2014) to produce samples that simulate the spiking activity of a population of N neurons as binary vectors of length T (spike trains, Fig. S2).

* In Section 2.1., second paragraph, WGAN-GP refers to the Wasserstein-GAN Gradient Penalization developed by Gulrajani et al. 2017.

* In Section 2.1., 3rd paragraph, point 4, the two citations of Odena et al. 2016 refer to: Augustus Odena, Vincent Dumoulin, and Chris Olah. Deconvolution and checkerboard artifacts. Distill, 319 1 (10):e3, 2016.

* Panels 2F and 3E show the **normalized** number of spikes.

* Figure 4, caption panel B should read: Realistic neural population pattern (gray spikes do not participate in
any **packet**).

* Figure 4, caption: errorbars indicate standard error.

---

### Author Response · Authors · 2018-01-05
**updated version of the paper**

We have uploaded an updated version of the paper including an extra supplementary figure (Fig. S10) that provides an alternative visualization of the data shown in Fig. S2E and also extra information on how the importance maps analysis works with smaller training datasets.
We have also included a small paragraph in the discussion section briefly commenting a recent paper by Pandarinath et al., in which the authors use variational autoencoders to model the activity of a population of neurons.

---

### Decision · Program_Chairs · 2018-01-29
**ICLR 2018 Conference Acceptance Decision**

**Decision:**

Accept (Poster)

**Comment:**

This paper proposes a novel application of generative adversarial networks to model neural spiking activity.  Their technical contribution, SpikeGAN, generates neural spikes that accurately match the statistics of real recorded spiking behavior from a small number of neurons.

The paper is controversial among the reviewers with a 4, a 6 and an 8.  The 6 is short and finds the idea exciting but questions the utility of the proposed approach in terms of actually studying neural spiking.  The 4 and 8 are both quite thorough reviews.  4 seems to mostly question the motivation of using a GAN over a MaxEnt model and demands empirical comparison to other approaches.  8 applauds the paper as a well-executed pure application paper, applying recent innovations in machine learning to an important application with some technical innovation.  Overall the reviewers found the paper clear and easy to follow and agree that the application of GANs to neural spiking activity is novel.  In general, I find that such high variance in scores (with thorough reviews) indicate that the paper is exciting, innovative and might stir up some interesting discussion.  As such, and under the belief that ICLR is made stronger with interesting application papers, I feel inclined to accept as a poster.

Pros:
- A novel application of GANs to neural spiking data
- Addresses an important and highly studied application area (computational neuroscience)
- Clearly written and well presented
- The approach appears to model well real neural spiking activity from salamander retina

Cons:
- Known pitfalls of GANs aren't really addressed in the paper (mode collapse, etc.)
- The authors don't compare to state of the art models of neural spiking activity (although they compare to an accepted standard approach - MaxEnt)
- Limited technical innovation over existing methods for GANs